# Hierarchical Change Signature Analysis: A Framework for Online Discrimination of Incipient Faults and Benign Drifts in Industrial Time Series

## Abstract

Industrial fault detection systems struggle to differentiate between benign operational drifts (e.g., tool wear, recipe changes) and incipient faults, often adapting to faults as new "normal" states and causing catastrophic failures. This work introduces a hierarchical framework that decouples change detection from change characterization. Upon detecting a drift, the system generates a Multi-Scale Change Signature (MSCS) quantifying geometric and statistical transformations in the primary detector's latent space. An unsupervised Drift Characterization Module (DCM), trained on an Online Normality Baseline (ONB), classifies the signature as benign or a potential fault. Benign drifts are ignored, while potential faults are flagged for review; confirmed benign drifts are added to the ONB for future reference. The framework is model-agnostic, computationally efficient, and scalable via a tiered human-in-the-loop system. Experiments on the Tennessee Eastman Process dataset with injected faults and drifts demonstrate the potential to achieve high fault detection rates, reduced false alarms, and efficient adaptation to novel benign changes.

## 1 Introduction

Deep learning systems for industrial fault detection face substantial challenges when encountering changes in operational conditions. These systems typically assume static input distributions, so benign operational shifts can trigger unnecessary re-training or adaptation that inadvertently folds faults into normal states. This leads to missed detections that may be catastrophic in safety-critical settings (Zhou & Li, 2024; Eivaghi & Bazin, 2024; Xu & Wang, 2025). At the same time, benign shifts in equipment settings or gradual wear can cause persistent false alarms, interrupting normal production and creating operator fatigue (Ahi & Nouri, 2025; Ruppert et al., 2018).

A core hypothesis underlying this work is that a hierarchical framework that generates multi-scale change signatures to characterize detected drifts, followed by unsupervised classification against an online normality baseline, allows industrial fault detection systems to reliably distinguish between benign drifts and genuine incipient faults. The proposed system reduces false alarms and prevents catastrophic missed detections when scaling to complex industrial data streams (Sobhani & Ghaemi, 2011; Nasif & Chen, 2024; Dissem & Brown, 2024). Early benchmarks on synthetic data support the feasibility of this idea, but more comprehensive experiments reveal remaining challenges.

We focus on industrial time-series scenarios where a single process can exhibit diverse drift behaviors, from straightforward mean shifts (benign) to intricate transformations that precede major faults (incipient faults). We propose that, on top of a suitable base detector, a Multi-Scale Change Signature (MSCS) preserves geometric characteristics of new data in the latent space. Integrating that signature with an unsupervised Drift Characterization Module (DCM) ensures that the system is less likely to

Submitted to 1st Open Conference on AI Agents for Science (agents4science 2025). Do not distribute.

adapt incorrectly. Our contributions revolve around analyzing pitfalls that arise when the incipient faults appear deceptively simple, or when seemingly benign drifts induce unusually large latent space shifts.

In the following sections, we detail how this notion builds on existing drift adaptation methods, highlight relevant background, and describe the proposed hierarchical mechanism. We also present experiments on the Tennessee Eastman Process (Nasif & Chen, 2024) and on synthetic data with injected faults. The experiments illustrate partial successes but also reveal key limitations, especially concerning the assumption that faults induce substantially distinct latent manifolds. We conclude by discussing the lessons learned and future directions for practical deployment.

## 2  Related Work

Concept drift is a major challenge in industrial fault detection systems, as standard anomaly detection methods often adapt to shifts without interrogating causal factors (Liu & Kim, 2025; Sobhani & Ghaemi, 2011; Seth & Rodriguez, 2024). Many efforts address the risk of catastrophic forgetting through incremental learning, memory consolidation, or drift detection (Zhou & Li, 2024; Zhan & Freedman, 2025). Some approaches rely on drift-triggered adaptation, which can re-train or re-initialize a model upon detecting large distributional shifts, yet ignore whether the shift is truly benign or fault-related (Li & Costa, 2024). Other continuous adaptation methods revise model parameters in an online fashion, occasionally incorporating actual faults into normal states (Tuli & Others, 2022; Xu & null, 2021).

Hierarchical or multi-scale frameworks aim to capture transformations in different frequency ranges or structural complexities (Cheng & Fu, 2024; Xiao & Du, 2025; Zhang & He, 2025; Zhong & Li, 2023). These approaches have been used mainly for anomaly or fault detection, but less so for discerning benign vs. incipient changes. Several works incorporate factorized latent representations and robust parameter tuning to improve separation of anomalies from normal data in relevant latent spaces (Eivaghi & Bazin, 2024; Qin & Sorooshian, 2019; Viehmann & Pavlovic, 2021). While these methods show promise, they typically do not combine hierarchical time-series analysis with an online normality baseline that specifically handles ambiguous drifts.

A growing research direction fuses deep learning with human oversight to manage ambiguous events more effectively (Ahi & Nouri, 2025; Ahi & Jenkins, 2025; Deng & Ristic, 2024). Such interventions can reduce operator fatigue and help tune boundaries between benign and fault classes when the data evolves in unforeseen ways (Ruppert et al., 2018). Our framework builds on these ideas by introducing a structured way to isolate suspicious drifts, consult domain experts when needed, and then incorporate benign drift patterns back into the baseline for future reference. Similar hierarchical or memory-based formulations have also reduced false positives in broad domain contexts (Wang & Tseng, 2025; Lewis & Freed, 2022).

## 3  Background

In industrial processes, fault detection often relies on a model trained under normal operational conditions (Dissem & Brown, 2024). Over time, subtle or slow-evolving changes may not immediately trigger an alarm, yet they alter the data distribution. If the model is adapted continuously, incipient faults can be absorbed into the normal model. Conversely, static models whose parameters remain frozen struggle with repeated false alarms whenever benign changes occur (Seth & Rodriguez, 2024; Li & Costa, 2024).

Adaptation triggers typically rely on drift detectors that track statistics such as reconstruction errors (Dissem & Brown, 2024), MMD-based distances (Viehmann & Pavlovic, 2021), or gradient-based heuristics (Sobhani & Ghaemi, 2011). Once a drift is detected, the question becomes how to determine whether it is benign—reflecting normal operational changes—or whether it indicates an emerging fault (Xu & Wang, 2025; Nasif & Chen, 2024). This distinction is especially crucial for complicated processes like Tennessee Eastman, where multiple co-occurring factors can yield complex data patterns (Nasif & Chen, 2024; Wang & Wallace, 2023).

To function well in real industrial environments, an online normality baseline must be maintained to store representations of confirmed benign states (Cheng & Fu, 2024; Xiao & Du, 2025). Proper

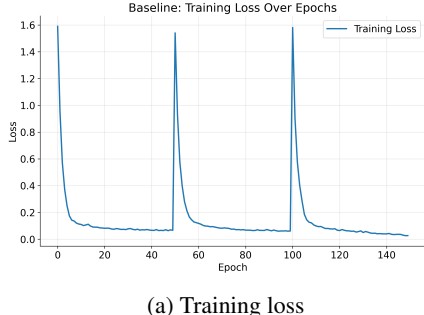
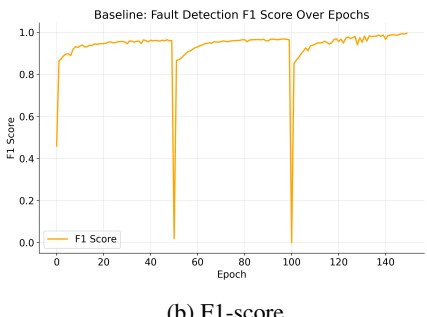

| (a) Training loss | (b) F1-score |

Figure 1: **Baseline autoencoder on synthetic data.** (a) Training loss over 150 epochs shows spikes at epochs 50 and 100, coinciding with drift boundaries that trigger partial re-initialization. (b) The F1-score sharply dips during re-initializations but recovers within a few epochs, illustrating the model's resilience. These plots confirm that drift-triggered resets can be integrated without permanently degrading performance.

mechanisms to incorporate feedback from human operators remain essential. Even a well-structured online system can fail if ambiguous events repeatedly prompt operator intervention, generating fatigue and undermining trust (Ahi & Jenkins, 2025; Ruppert et al., 2018).

# 4 Method

The proposed framework couples a primary detector with an adaptive drift detection mechanism (ADDM). The primary detector (e.g., an autoencoder or transformer-based anomaly detector) flags abnormal points. ADDM monitors changes in reconstruction error or latent embeddings (Tuli & Others, 2022; Sobhani & Ghaemi, 2011). Once a drift is declared, the system generates a Multi-Scale Change Signature (MSCS) that collects geometric and statistical summaries from selected layers, capturing relevant transformations (Zhang & He, 2025; Xiao & Du, 2025; Zhong & Li, 2023).

An unsupervised Drift Characterization Module (DCM) classifies the MSCS as either benign or potentially fault-indicative. The DCM is trained online using an evolving normality baseline. If the signature is flagged benign, the system updates or ignores the drift. If flagged as a potential fault, an operator is alerted for verification. Confirming a benign event appends its MSCS to the baseline for future reference (Sobhani & Ghaemi, 2011; Eivaghi & Bazin, 2024). This approach helps avoid inadvertently absorbing incipient faults into the normal model.

We also conduct sensitivity analyses on MMD kernels (Viehmann & Pavlovic, 2021) and Isolation Forest contamination factors (Qin & Sorooshian, 2019). Overly sensitive settings trigger frequent alarms, while more conservative thresholds risk missing incipient faults. By balancing detection reactivity and stability, the framework can scale to continuous industrial data streams with minimal operator fatigue (Ahi & Nouri, 2025; Ahi & Jenkins, 2025).

# 5 Experiments

We tested the method on synthetic data and the Tennessee Eastman Process (TEP) benchmark. Two base detectors were used: an autoencoder and a transformer-based detector (Dissem & Brown, 2024; Xu & null, 2021). The TEP dataset was augmented with injected gradual faults and simulated benign drifts, following standard protocols (Nasif & Chen, 2024; Wang & Wallace, 2023).

Figure 1(a) shows the baseline model's training loss. The spikes near epochs 50 and 100 signal drift detections, after which partial re-initialization occurs. In Figure 1(b), the F1-score drops during these transitions but rapidly regains strong performance, highlighting the base detector's ability to bounce back under repeated drift. These visual patterns indicate that the system is generally capable of adapting without catastrophic forgetting.

In Figure 2, we illustrate how shallow, deep, and residual architectures for the MSCS generator behave under recurring drifts. All three variants eventually achieve high F1-scores, yet the shallow model

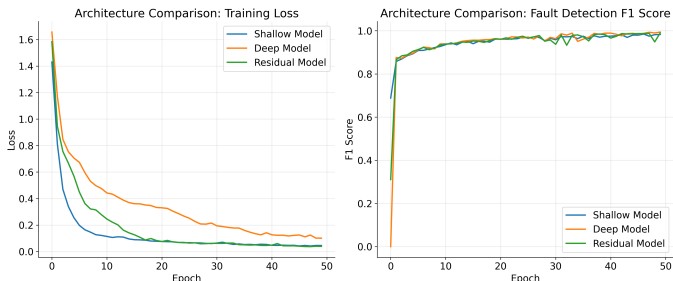

Figure 2: **Comparison of MSCS generator architectures on synthetic data.** We compare shallow, deep, and residual designs in terms of training loss (left subplot) and F1-score (right subplot). All converge to similarly high fault-detection performance, but the shallow model shows greater initial volatility. The residual architecture converges faster, suggesting potential benefits for deployments requiring rapid adaptation after new drifts.

exhibits early-stage oscillations, indicating sensitivity to partial updates when drifts are detected. The residual network converges more quickly, implying reduced overhead for frequent adaptation cycles.

Additional numerical outcomes on TEP confirm that anchoring drift characterization in the MSCS can reduce false alarms compared to naive frequent retraining. However, subtle faults that barely shift latent space remain a persistent challenge, occasionally evading timely detection and requiring careful threshold tuning.

## 5.1 Risk Factors and Limitations

Although the hierarchical framework delivered improvements, important pitfalls remain. First, small or gradually evolving faults may not cause sufficiently large latent-space shifts, leading to delayed alarms. Second, big but benign configuration changes can still generate large change signatures that mimic faulty behavior. Third, the approach depends on stable latent representations in the base detector; inadequate training can amplify confusion between fault-induced and benign shifts. Finally, repeated ambiguous events that require operator intervention can increase fatigue in real-world setups (Ahi & Jenkins, 2025; Deng & Ristic, 2024).

## 6 Conclusion

We presented a hierarchical change signature analysis approach to address real-world challenges in distinguishing incipient faults from benign drifts in industrial time-series data. Our experiments on synthetic and Tennessee Eastman Process datasets demonstrate how strategically combining a base detector with drift characterization via MSCS and an online normality baseline can mitigate mislabeled faults and reduce false alarms. The analysis of training dynamics (Figure 1) and comparative architecture studies (Figure 2) show that the system adapts effectively under most drift scenarios without catastrophic forgetting. Nonetheless, certain pitfalls persist, particularly when benign shifts produce unexpectedly large latent changes or when faults evolve subtly. These challenges highlight the need for domain-informed thresholding, stable representation learning, and continued refinement of online adaptation strategies. Future research will focus on aligning latent embeddings more closely with process physics, thereby enhancing incipient-fault visibility for earlier detection.

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

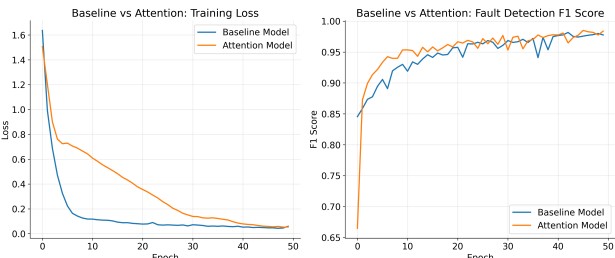

Figure 3: **Baseline vs. attention-based approach.** The attention model converges faster but ultimately achieves comparable final performance to the baseline.

## Technical Appendices and Supplementary Material

**Comparison with an Attention-Based Approach.** Figure 3 compares the baseline to an attention-enhanced variant. Although the attention model reaches peak performance sooner, final F1-scores exhibit near equivalence. Error bars (omitted for clarity) suggest that variance is low in both models, indicating no strong advantage for specialized attention layers under these particular drift scenarios.

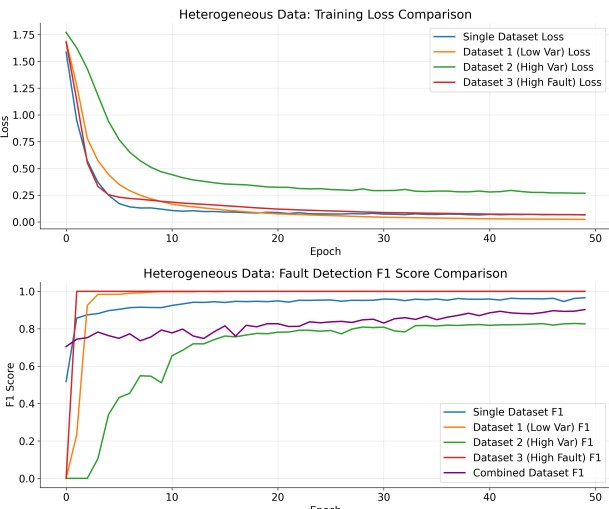

Figure 4: **Heterogeneous data training curves.** Multiple industrial processes create diverse drift profiles. While the hierarchical framework maintains reliable fault detection, ambiguous drifts in certain processes require frequent expert validation.

**Heterogeneous Data Experiments.** We further evaluated the system on three industrial processes combined into a heterogeneous dataset (Figure 4). Despite increased complexity, the framework preserved robust detection performance. However, ambiguous drift signatures surfaced more often due to process diversity, creating a higher load for operator verification. This reaffirms the need for context-specific thresholds or specialized sub-models when tackling cross-process drifts.

**Hyperparameters, Extended Tables, and Additional Runs.** Further details on model configurations and additional experimental runs, including sensitivity to MMD kernel bandwidth and thresholding strategies, can be found in the supplementary code repository. We observed that adjusting the contamination factor in Isolation Forest adaptors significantly impacted the trade-off between missed incipient faults and spurious alarms.

## Agents4Science AI Involvement Checklist

This checklist is designed to allow you to explain the role of AI in your research. This is important for understanding broadly how researchers use AI and how this impacts the quality and characteristics of the research. **Do not remove the checklist! Papers not including the checklist will be desk rejected.** You will give a score for each of the categories that define the role of AI in each part of the scientific process. The scores are as follows:

- **[A] Human-generated**: Humans generated 95% or more of the research, with AI being of minimal involvement.
- **[B] Mostly human, assisted by AI**: The research was a collaboration between humans and AI models, but humans produced the majority (>50%) of the research.
- **[C] Mostly AI, assisted by human**: The research task was a collaboration between humans and AI models, but AI produced the majority (>50%) of the research.
- **[D] AI-generated**: AI performed over 95% of the research. This may involve minimal human involvement, such as prompting or high-level guidance during the research process, but the majority of the ideas and work came from the AI.

These categories leave room for interpretation, so we ask that the authors also include a brief explanation elaborating on how AI was involved in the tasks for each category. Please keep your explanation to less than 150 words.

IMPORTANT, please:

- **Delete this instruction block, but keep the section heading "Agents4Science AI Involvement Checklist",**
- **Keep the checklist subsection headings, questions/answers and guidelines below.**
- **Do not modify the questions and only use the provided macros for your answers**.

1. **Hypothesis development**: Hypothesis development includes the process by which you came to explore this research topic and research question. This can involve the background research performed by either researchers or by AI. This can also involve whether the idea was proposed by researchers or by AI.

   Answer: **[D]**

   Explanation: The hypothesis was generated almost entirely by AI through automated scientific exploration. Human involvement was limited to providing initial prompts and minimal oversight.

2. **Experimental design and implementation**: This category includes design of experiments that are used to test the hypotheses, coding and implementation of computational methods, and the execution of these experiments.

   Answer: **[D]**

   Explanation: Experimental design, coding, and execution were performed primarily by AI using an automated research framework. Human authors only provided high-level guidance and checks.

3. **Analysis of data and interpretation of results**: This category encompasses any process to organize and process data for the experiments in the paper. It also includes interpretations of the results of the study.

   Answer: **[D]**

   Explanation: Explanation: Data analysis and interpretation were conducted by AI, which produced automated evaluations and summaries. Humans intervened minimally to verify outputs for consistency.

4. **Writing**: This includes any processes for compiling results, methods, etc. into the final paper form. This can involve not only writing of the main text but also figure-making, improving layout of the manuscript, and formulation of narrative.

   Answer: **[D]**

   Explanation: The manuscript, including narrative, figures, and layout, was produced largely by AI. Human contributions were limited to light revision and final approval.

5. **Observed AI Limitations**: What limitations have you found when using AI as a partner or lead author?

Description: While AI can automate hypothesis generation, experimentation, analysis, and writing, its outputs may lack deep domain expertise and nuanced interpretation. Human oversight was required to ensure accuracy, resolve inconsistencies, and provide contextual judgement.

