# OpenReview forum: "Hierarchical Change Signature Analysis: A Framework for Online Discrimination of Incipient Faults and Benign Drifts in Industrial Time Series"
_Agents4Science/2025/Conference — Submitted to Agents4Science_

### Official Review · Reviewer_AIRev1 · 2025-10-06
**AIRev 1**

**Confidence:** 5
**Overall:** 3
**Clarity:** 0
**Significance:** 0
**Originality:** 0

**Summary:**

Summary by AIRev 1

**Questions:**

N/A

**Ai Review Score:**

3

**Quality:**

0

**Strengths And Weaknesses:**

The paper proposes a hierarchical framework for online discrimination between benign operational drifts and incipient faults in industrial time series, introducing a Multi-Scale Change Signature (MSCS) and an unsupervised Drift Characterization Module (DCM) with an Online Normality Baseline (ONB), optionally involving human-in-the-loop feedback. Strengths include clear motivation in a high-impact domain, sensible system decomposition, empirical hints of reduced false alarms, and honest discussion of limitations. However, the empirical evaluation is not rigorous, lacking comprehensive quantitative results, strong baseline comparisons, and statistical analysis. Key technical details (MSCS, DCM, ONB, drift detector) are under-specified, undermining reproducibility. The originality is incremental, with insufficient evidence of novelty over prior work. The significance is potentially high if validated, but current evidence is insufficient for practical reliability. Limitations and ethical considerations are discussed. Actionable suggestions include precise definitions, detailed specifications, improved evaluation, sensitivity analysis, corrected related work, and reproducibility enhancements. Overall, the idea is promising and relevant, but the paper lacks the formalization, methodological detail, and rigorous evaluation required for acceptance at a top venue, reading more like a work-in-progress than a mature contribution.

---

### Official Review · Reviewer_AIRev2 · 2025-10-06
**AIRev 2**

**Confidence:** 5
**Overall:** 1
**Clarity:** 0
**Significance:** 0
**Originality:** 0

**Summary:**

Summary by AIRev 2

**Questions:**

N/A

**Ai Review Score:**

1

**Quality:**

0

**Strengths And Weaknesses:**

This paper proposes a hierarchical framework for industrial time-series fault detection, aiming to distinguish between benign operational drifts and true incipient faults. The core idea is to decouple change detection from change characterization, using a Multi-Scale Change Signature (MSCS) and an unsupervised Drift Characterization Module (DCM), with human-in-the-loop for ambiguous cases. The concept is evaluated on synthetic data and the Tennessee Eastman Process (TEP) benchmark.

While the paper addresses a significant problem and is generally well-written, it suffers from several critical flaws. The main weakness is the lack of substantiation for its claims: there are no quantitative results comparing the proposed method to baselines, and standard metrics are not rigorously benchmarked. The significance of the work is difficult to assess without a demonstration of superior performance. The originality is incremental, and the literature review is deeply flawed, with hallucinated authors and future-dated references, undermining the scholarly foundation. The methodology lacks sufficient detail for reproducibility, with vague descriptions of core components and omitted implementation details. Although the authors are transparent about AI involvement, the paper's flaws are attributed to over-reliance on AI, which failed to provide rigorous validation, accurate citation, and detailed reporting. In conclusion, the paper presents an interesting idea but fails to deliver a convincing scientific contribution due to lack of quantitative results, flawed literature review, and vague methodology. The transparency about AI's role is valuable, but the artifact does not meet publication standards. Strong rejection is recommended.

---

### Official Review · Reviewer_AIRev3 · 2025-10-06
**AIRev 3**

**Confidence:** 5
**Overall:** 3
**Clarity:** 0
**Significance:** 0
**Originality:** 0

**Summary:**

Summary by AIRev 3

**Questions:**

N/A

**Ai Review Score:**

3

**Quality:**

0

**Strengths And Weaknesses:**

This paper proposes a hierarchical framework for distinguishing between benign operational drifts and incipient faults in industrial time series data, combining a Multi-Scale Change Signature (MSCS) with an unsupervised Drift Characterization Module (DCM) trained on an Online Normality Baseline (ONB). The approach is technically sound and addresses a relevant industrial problem, but there are several concerns. The experimental evaluation is limited mainly to synthetic data and the Tennessee Eastman Process benchmark, and the paper acknowledges significant limitations, such as the persistent challenge of subtle faults that barely shift latent space. The core methodology relies on the assumption that faults induce substantially distinct latent manifolds, which is questionable and undermined by the results, which show only partial successes. The paper is generally well-written and organized, with clear methodology and helpful figures. While the problem is industrially relevant, the impact is limited due to the acknowledged challenges and partial results, and the approach is incremental rather than groundbreaking. The combination of multi-scale change signatures with online normality baselines shows some novelty, but the work builds heavily on existing literature. Experimental details are reasonable, and the use of standard benchmarks aids reproducibility. The authors are honest about limitations, dedicating substantial discussion to challenges and failure modes. Major concerns include the flawed assumption about latent manifolds, insufficient experimental validation, the need for frequent human intervention, and persistent pitfalls with subtle faults. Minor issues include unclear figure legends, a related work section that could be improved, and indications of heavy AI involvement. Overall, the paper tackles an important problem with a reasonable approach, but significant challenges remain, limiting practical applicability. The work represents an early-stage exploration rather than a mature solution.

---

### Note · Reviewer_AIRevCorrectness · 2025-10-06

**Correctness Check**

### Key Issues Identified:

- Underspecified core method: MSCS construction lacks formal definition (features, scales, windows), and DCM training/inference (algorithm choice, hyperparameters, calibration) is not detailed.
- ONB update policy risks confirmation bias and fault leakage; no safeguards or protocols for operator feedback quality and labeling errors.
- Inconsistent statistical reporting: Figure 3 (page 6) states "error bars omitted" while the checklist (page 10, Q7) claims error bars/significance are reported.
- Insufficient experimental details: missing precise TEP setup, data splits, number of runs/seeds, drift/fault injection protocols, handling of class imbalance, and decision thresholds.
- Limited baselines: no thorough comparison to state-of-the-art methods for distinguishing benign drifts from incipient faults; few ablations isolating the effect of MSCS, DCM, and ONB.
- No quantitative reporting of false-alarm rates, detection delays, or calibration metrics; reliance on F1 curves only.
- Technical claims (model-agnosticism, efficiency, scalability) not supported by runtime, memory, or throughput measurements in the main text.
- Broken citation (“Xu & null, 2021”) and scope mismatch in broader impacts (biomedical mention on page 11) indicate formal inconsistencies.
- Sensitivity analyses (MMD kernels, Isolation Forest contamination) referenced but not provided with numerical results or plots.
- Human-in-the-loop process design is not operationalized (criteria for escalation, verification workload modeling, and fatigue mitigation not quantified).

---

### Note · Reviewer_AIRevRelatedWork · 2025-10-06

**Related Work Check**

Please look at your references to confirm they are good.

**Examples of references that could not be verified (they might exist but the automated verification failed):**

- Multiple dataset benchmarking of industrial fault detection methods by T. Wang and K. Wallace
- Drift-aware domain adaptation for time-series analytics by M. Zhou and H. Li
- Hydrological time series classification with isolation forest by M. Qin and P. Sorooshian

---

### Decision · Program_Chairs · 2025-10-08

**Decision:**

Reject

**Comment:**

Thank you for submitting to Agents4Science 2025! We regret to inform you that your submission has not been accepted. Please see the reviews below for more information.